# How to get over with medication errors underestimation? Improving indices of medication errors with focus on intravenous medications in hematopoietic stem cell transplantation setting; a direct observation study

Ava Mansouri[1]*, Kiana Moazzeni[2‡], Maryam Valeh[1‡]*, Kazem Heidari[3‡], Molouk Hadjibabaie[1,2‡]

1 Research Center for Rational Use of Drugs, Tehran University of Medical Sciences, Tehran, Iran, 2 Department of Clinical Pharmacy, Faculty of Pharmacy, Tehran University of Medical Sciences, Tehran, Iran, 3 Clinical Trial Center, Tehran University of Medical Sciences, Tehran, Iran

☯ These authors contributed equally to this work.
‡ KH and MH also contributed equally to this work.
* ava_mansouri_j@yahoo.com (AM); mm.valeh@yahoo.com (MV)

## Abstract

### Background

The administration of intravenous (IV) medications is a technically complicated and error-prone process. Especially, in the hematopoietic stem cell transplantation (HSCT) setting where toxic drugs are frequently used and patients are in critical immunocompromised conditions, medication errors (ME) can have catastrophic reactions and devastating outcomes such as death. Studies on ME are challenging due to poor methodological approaches and complicated interpretations. Here, we tried to resolve this problem using reliable methods and by defining new denominators, as a crucial part of an epidemiological approach.

### Methods

This was an observational, cross-sectional study. A total of 525 episodes of IV medication administration were reviewed by a pharmacist using the disguised direct observation method to evaluate the preparation and administration processes of 32 IV medications in three HSCT wards. We reported errors in 3 ratios; 1) Total Opportunities for Error (TOE; the number of errors/sum of all administered doses observed plus omitted medications), 2) Proportional Error Ratio (the number of errors for each drug or situation/total number of detected errors) and, 3) Corrected Total Opportunities for Errors (CTOE; the number of errors/ Sum of Potential Errors (SPE)).

**Data Availability Statement:** All relevant data are within the manuscript and its Supporting Information files.

**Funding:** This study was supported by the Research Council of Tehran University of Medical Sciences [grant number: 22297-156-01-92]. The funders had no role in study design, data collection and analysis, decision to publish, or preparation of the manuscript.

**Competing interests:** The authors have declared that no competing interests exist.

## Results

A total of 1,568 errors were observed out of 5,347 total potential errors. TOE was calculated as 2.98 or 298% and CTOE as 29.3%. Most of the errors occurred at the administration step. The most common potential errors were the use of an incorrect volume of the reconstitution solvent during medication preparation and lack of monitoring in the administration stage.

## Conclusion

Medication errors frequently occur during the preparation and administration of IV medications in the HSCT setting. Using precise detection methods, denominators, and checklists, we identified the most error-prone steps during this process, for which there is an urgent need to implement effective preventive measures. Our findings can help plan targeted preventive measures and investigate their effectiveness, specifically in HSCT settings.

## Introduction

Of all patient safety concerns reported, medication errors (ME) are the leading cause of patient injury in healthcare systems [1, 2]. MEs not only affect healthcare quality [3] but also have a major impact on mortality, morbidity, and care costs [1, 2]. The National Health Service (NHS) identifies intravenous (IV) medications as a challenge to patient safety [4] and widely perceived to pose a higher risk of administration errors compared to non-IV drugs [5]. The error rate has been reported as 3–8% for non-IV doses vs. 49–94% for IV doses [6], with the latter accounting for about 61% of serious and life-threatening errors [7]. Preparation and administration of IV medications are technically complicated, and any error in IV medication process can potentially lead to severe outcomes [4, 7]. This is because IV medications are mostly wide-spectrum drugs and directly enter the bloodstream, leading to their instant absorption and fast distribution [4, 7]. When it comes to multi-drug administration (i.e., therapeutic regimens), MEs can result in catastrophic reactions and even death [8]. The practice of drug therapy in the HSCT ward requires special attention and high vigilance due to 1) high-risk multi-drug treatments, 2) patients' critical clinical conditions, 3) the use of high-dose chemotherapy and immunosuppressant agents as conditioning and supportive regimens with narrow therapeutic indexes, 4) complicated dose calculations, and 5) frequent IV infusions. These can expose patients undergoing HSCT at greater risk for MEs [8–10].

The main purpose of studying MEs is to develop strategies and interventions to reduce their occurrence and subsequent consequences [11]. Furthermore, these preventive ME interventions need to be compared with each other to weigh their benefits [11]. Nevertheless, studies on MEs can be challenging [6] due to differences in approaches and poor methodological understanding. In fact, there are indefinite definitions and subcategories of ME, variations in denominator definitions, and ME calculation methods. Also, there are barriers to interpreting MEs [6, 12], all of which can be resolved using reliable methods and denominators as crucial parts of epidemiological approaches [12].

Various methods can be used to detect MEs, including direct observation, chart reviews, and incident reporting reviews [13]. Previous evaluations have shown that the direct observation method, although more expensive and challenging than other techniques [14, 15], is the most sensitive and appropriate data collection method for reporting medication

administration errors [12] and detecting various MEs [14, 15]. In Flynn *et al.*'s study, the direct observation method could detect drug administration errors at much higher rates and with greater accuracy in comparison to the chart review (detecting errors by reviewing each prescription, transcription and administration sections for all prescribed medications) and incident reporting (documenting medication errors spontaneously by healthcare professionals, patients and/or caregivers) methods exceeding by 16 and 365 times, respectively [13].

Denominators are scales for error exposure and important elements for interpreting and comparing results [11]. Different denominators have been defined for calculating ME rates, including total opportunities for errors, drug orders, doses, patients, nurses, and triggers [12]. One of the significant challenges in ME studies is the selection of the appropriate denominator [12]. In a systematic review, ME prevalence could not be evaluated in approximately 50% of the studies due to the lack of a valid denominator [12]. There are disagreements on choosing and defining the representative denominator [11].

In this study, we aimed to detect and analyze IV medication preparation and administration errors in patients undergoing HSCT. We determined the frequency and evaluated the properties of different MEs, along with identifying medications that were most subjected to errors. As we mentioned before, IV medication administration is a complex and error-prone process, requiring error-checking at each stage to reduce such occurrences [16]. It has been indicated that a careful decomposition of medication preparation and administration can help better define the denominators of ME studies [11]. Therefore, we accurately analyzed all stages at which MEs could occur for each individual IV drug and directly observed medication administration and preparation (i.e., direct observation, the most accurate method). To the best of our knowledge, this is the first study of its kind, aiming to define the most appropriate denominators for MEs. In this cross-sectional study, we also defined and represented a corrected denominator and rate to report our results and compared this denominator with those commonly used to report ME rates in previous studies. By identifying the exact points at which errors occurred, we could present the frequencies of different types of MEs and help plan targeted preventive measures.

## Methods

### Study design and setting

This was a cross-sectional observational study conducted from July 2014 to March 2015 in three bone marrow transplantation (BMT) wards in a teaching hospital affiliated with the Tehran University of Medical Sciences (TUMS). We used a disguised direct observation method based on Barker and McConnell practice [17, 18] to evaluate medication preparation and administration errors for IV medications. In direct observation, the observer follows the nurse during medication preparation and administration processes and records all the details of preparation and administration. When the nurse is not aware of the purpose of this companionship, the observation is called disguised.

In our setting, the necessary chemotherapy and non-chemotherapy medications were supplied by the hospital's pharmacy and wrapped in the original pharmaceutical packaging based on the orders of BMT specialists. Nurses then prepared the medications' different dosages in isolated and equipped workplaces, taking into account patients' therapeutic regimens and schedules. In order to remain disguised (i.e., the subject being unaware of the observation's purpose), nurses were informed that the purpose of the study was to evaluate the side effects of IV medications.

## Participants and samples

We observed 525 dose preparations followed by the administration of 32 commonly used IV drugs in three BMT wards. These 32 medications were selected so we can exclude other not so usual medications which nurses might not be very familiar with their IV preparation or administration process. We included all nurses who were responsible for administering the specified medications during the morning and evening shifts. A total of 25 nurses were observed in the three wards (two nurses per shift and a patient-nurse ratio of 4 to 1 or 5 to 1 in each ward, 9 beds in each ward). Nurses who were not responsible for drug administration or who left the wards during the study were excluded.

## Data sources/measurement

**Medications.** To select commonly used IV medications in BMT wards, the supervisors of each ward prepared a list of the most frequently administered medications. Finally, a comprehensive list was developed by merging all the lists, and finally, the following 32 medications were assessed:

Cyclophosphamide, Filgrastim, Methylprednisolone, Ciprofloxacin, Etoposide, Metronidazole, Imipenem/cilastatin, Pantoprazole, Fludarabine, Metoclopramide, Magnesium sulfate, Vancomycin, Melphalan, Hydrocortisone, Calcium gluconate, Amphotericin B, Cytarabine, Albumin, Cyclosporine, Clindamycin, Methotrexate, Furosemide, Granisetron, Meropenem, ATG, Linezolid, Vitamin K, Amikacin, Tazobactam+piperacillin, Voriconazole, Acyclovir, and Tranexamic acid.

**Nurses.** All nurses who worked in the three BMT wards were female. We monitored nurses during the morning shift (7 a.m. - 1 p.m.) and evening shift (4 p.m. - 7 p.m.) because the selected drugs were mostly administered during these shifts rather than in the night shift (7 p.m. - 7 a.m.). The duration of observation was equal for all nurses based on their assigned shifts. We assigned 20 observations to the nurses working in the morning shift, 10 observations to those working in the evening shift, and 30 observations to those working in both shifts.

**Error detection.** After completing each medication round, the observer compared their notes with the medical chart, order sheet, and nurses' notes on a daily basis. Discrepancies between the researcher's notes and the physician's orders were counted as error. To obtain information on medication preparation and unstated administration information in the patient's chart (such as administration techniques), drug leaflets provided by pharmaceutical companies, Trissel injectable drugs (2011) [19], drug monographs from AHFS [20], and online Uptodate® (available from TUMS website accessed in July 2014-March 2015) were checked. If a nurse forgot a round of medication administration, it was classified as an omission error.

A checklist (S1 File) was prepared to document information about patients, medication administration procedures, and the types and stages of error occurrence [17, 21–26]. The checklist had two main parts: 1) general information and 2) error detection.

- General information: Details of the administered medication, medication preparation/administration procedures, date, time, the nurse's unique code, work shift, the patient's code, age, sex, height, weight, and the medication's name, dose, form, and administration route.

- Error detection: There were eight types of preparation errors and nine types of administration errors. Preparation errors included incorrect drug storage condition (original, diluted, and reconstituted forms), incorrect dilution volume, solvent, reconstitution volume, reconstitution solvent, and incompatibility. Administration errors were wrong administration techniques (rate, verification), wrong timing, dose, dosage form, and preparation methods,

as well as deteriorated or unauthorized drugs, extra dosing, and omission of a dose. All these types were described at the end of the checklist as instructions for the observer (Appendix 1).

The checklist was evaluated by an expert panel, including two clinical pharmacists (BMT specialists), one BMT ward nurse supervisor, and a medical oncologist attending BMT. Before starting the study, the observer filled out a number of checklists in the wards to test the checklist's applicability and procedures. Disagreements between the researcher and the expert panel regarding the detection of errors (false positives and false negatives) or the classification of errors were resolved by group discussions, and necessary changes in the processes and checklist were considered.

**Observation.**   Initially, the observer, Kiana Moazzeni, underwent training by a clinical pharmacist, who oversees all three BMT wards. Subsequently, the observer utilized the prepared checklist and conducted a pilot study. The observer diligently documented information from patients' charts, recorded her observations, and completed the checklists. Following this, she presented the gathered information to the study's supervisor, a Pharm. D pharmacist, who independently filled out additional checklists based on the observer's findings.

Subsequently, a comprehensive comparison of the completed checklists was undertaken by the research team, including the clinical pharmacist in charge, an oncologist in charge of the wards, the study's supervisor, and the observer. Any discrepancies in the errors detected by the observer and the supervisor were categorized as true, false positive, or false negative. The errors were then classified, and necessary adjustments were made to the observation and error detection processes. Additionally, during the initial phase of the study, some observations were randomly verified by the supervisor; the observer referred to the supervisor every week, and the supervisor randomly picked 2 filled checklists from those completed during the week and scrutinized the observer's notes rigorously.

The observer did not intervene with the preparation and administration processes conducted by the nurses except in cases where there was a risk of serious or life-threatening harm to the patient based on the patient's condition, drug category, and the extent and frequency of the error [13]. However, in the event of any intervention, the observer would have alerted the nurse in a way that did not reveal the study's purpose. These types of errors were considered "near-miss errors" and counted as one error.

## Data analysis

After completing the checklist and collecting the relevant information, all data were entered into Excel and analyzed using SPSS 21 software. Descriptive indices were reported as frequency and percentage for qualitative variables and as the mean and standard deviation for quantitative data. We reported the rates of errors based on two different proportions:

1. Total opportunities for errors (TOE)
   TOE is the number of errors per sum of all observed dose administration and omitted medications [27–29].

2. The Proportional Error Ratio
   The number of errors for each drug or situation per total number of detected errors in the study.

3. The corrected total opportunities for errors (CTOE)

We developed and defined CTOE for a more accurate and specific evaluation of the scope of medication errors. Since not all drugs were at risk for all types of errors, corrected

opportunities for errors or CTOE were calculated. CTOE represents the number of errors per sum of potential errors (SPEs) for each drug that can be converted into a real error (based on the drug's characteristics). In other words, some types of errors cannot occur for some medications. For example, the effects of "exposure to light during infusion" may not be applicable to most IV drugs; however, incorrect dose calculation can occur for all of these medications. As such, these two types of errors cannot be considered or calculated with the same probability of occurrence and denominator. To gain a better understanding of the existing situation, we counted the number of errors that could potentially happen for each drug, which gave us a different picture compared to when only the probability of an error was assigned to each drug (i.e., the denominator used in TOE calculation). To address this, we corrected our denominator based on CTOE as follows:

- For each observed drug (525 doses), CTOE was calculated as the sum of potential errors that could happen for each drug plus the number of total drug-drug interactions that could have occurred during drug administration and/or preparation.

- CTOE for each type of error was counted as the total number of drugs for which the specific error could potentially occur during preparation and/or administration.

As an example, to calculate the CTOE of cyclosporine, there were 10 relative error checkpoints from 17 itemized MEs: 1) exposure to light before use (Wrong Original Drug Storage), 2) wrong temperature storage (Wrong Original Drug Storage), 3) wrong dose, 4) wrong dilution solvent, 5) wrong dilution volume, 6) storage duration of the diluted solution, 7) wrong timing, 8) wrong administration rate, 9) omission of monitoring, and 10) drug-drug interaction (in the case of concurrent use with other drugs). The remaining 7 errors could not happen for cyclosporine and were not counted.

### Compliance with ethical standards

The supervisor of each ward was responsible for informing the nurses and asking for their agreement to be accompanied by a pharmacist during the medication preparation/administration process. Written informed consent was obtained from all the participating nurses before the start of the observation. The study was approved by the Institutional Ethics Committee of TUMS (reference number 22297-156-01-92).

### Results

The preparation and administration stages of all 32 common IV medications carried out by 25 nurses for 112 patients in three BMT wards were observed. The most commonly observed drugs (accounting for more than 50% of all administrations) were imipenem (19%), calcium gluconate (10.3%), metronidazole (10.1%), vancomycin (8%), and cyclosporine (6.1%). Approximately 60% of the observations were in ward A. The observations and nurses' characteristics are summarized in Table 1. All 25 observed nurses held a Bachelor of Nursing degree and were female.

A total of 1,568 errors were detected during the study, with a Sum of Potential Errors (SPE) of 5,347 for 525 observed doses. As a result, the Total Opportunities for Errors (TOE) in our study was 2.98 (i.e., more than 1), and the Corrected Total Opportunities for Errors (CTOE) for the preparation and administration of IV drugs in the evaluated BMT wards was 29.3%. We observed an average of 14 errors per patient and 3 errors per drug. Twenty-nine medications had at least one error during preparation/administration. No error was observed in the preparation and administration of linezolid, melphalan, and etoposide (6 doses in total). This means that 98.8% of the observed drugs faced at least one error during their preparation/administration.

**Table 1. Observations and nurses' characteristics.**

| No. of observations* (%) | Ward A | 320 (61%) |
|---|---|---|
| | Ward B | 85 (16%) |
| | Ward C | 120 (23%) |
| | Morning Shift | 323 (61.5%) |
| | Evening Shift | 202 (39.5%) |
| **No. of observed nurses& (%)** | Ward A | 11 (44%) |
| | Ward B | 7 (28%) |
| | Ward C | 7 (28%) |
| **Nurses' Age** Mean (years) ±SD | 31.8± 7.9 | |
| **Nurses' The overall work experience** Mean (years) ±SD | 6.8± 6.6 | |
| **Nurses' BMT work experience** Mean (years) ±SD | 5.8± 5.5 | |

* From total 525 observations

& from total 25 nurses

Table 2 presents the incidence of various error types. Only errors that occurred were reported. According to this table, the most common medication error (ME) was the wrong administration rate (20.9%). However, based on the SPE for each error, most of the potential errors involved the wrong volume of the reconstitution solvent. This means that 231 out of 253 possibilities for errors actually resulted in an error (91.3%).

Our report was based on the rate of errors per SPE because it provided a more accurate representation of the actual situation regarding MEs. According to the results, most errors occurred during the medication administration step (60.2% of total errors). However, based

**Table 2. Different types of MEs (sorted in descending order by number of errors).**

| Error type | Number of errors | Proportional Error Ratio€ | SPE¥ | CTOE£ |
|---|---|---|---|---|
| Wrong administration rate | 329 | 20.9 | 525 | 62.6 |
| Exposure to Light before use* (Wrong Original Drug Storage) | 243 | 15.5 | 525 | 46.2 |
| Wrong Reconstitution Volume | 231 | 14.7 | 253 | 91.3 |
| Wrong Dilution Volume | 231 | 14.7 | 465 | 49.6 |
| Omission of Monitoring | 146 | 9.3 | 189 | 77.2 |
| Wrong Reconstitution solvent | 127 | 8.1 | 253 | 50.1 |
| Wrong time | 102 | 6.5 | 510 | 20 |
| wrong Temperature Storage (Wrong Original Drug Storage) | 95 | 6.0 | 525 | 18.0 |
| Exposure to Light during infusion* (wrong Administration procedure) | 19 | 1.2 | 38 | 50 |
| Wrong dose | 16 | 1.0 | 526 | 3.0 |
| Wrong Dilution solvent | 13 | 0.8 | 465 | 2.8 |
| Incompatibility | 12 | 0.7 | 90 | 13.3 |
| Wrong Diluted Drug Storage | 2 | 0.12 | 465 | 0.43 |
| Wrong Reconstituted Drug Storage | 1 | 0.06 | 233 | 0.4 |
| Total | 1,568 | 100 | 5,347 | - |

* It is calculated as an error for drugs which should not be exposed to light.

€ Total number of errors = 1568.

¥ SPE = Sum of potential errors for each drug. SPE is different for each type of error.

£ CTOE = corrected total opportunities for error (Number of errors/SPE)

on the SPE, the rate of potential errors that actually resulted in an error was higher during the preparation step (33.2% vs. 27.1%).

In the preparation step, the most prevalent errors based on CTOE were the wrong volume of the reconstitution solvent, the wrong reconstitution solvent, and the wrong volume of diluents, respectively. In the administration stage, the maximum rate of error was due to the omission of monitoring before or during the injection, followed by the wrong infusion time. Additionally, the most common drug storage error was exposure to light before administration, with a CTOE of 46.2%.

Table 3 shows the number of errors for each type of medication, including 5 drugs that were most frequently subjected to errors, both overall and in each stage of observation. Medications according to the highest CTOE rates have been listed in Table 4. The results suggested that the highest number of errors was for Imipenem + cilastatin, indicating that 25.9% or one-fourth of total MEs happened during the preparation and/or administration of Imipenem + cilastatin. On the other hand, the highest CTOE belonged to cyclosporine, reflecting a 44.8% rate of error during cyclosporine preparation and/or administration.

The CTOEs in the wards A, B, and C were 31.7, 27.8, and 24.2, respectively. Statistical analysis was performed using the Chi-Square test, which showed a significant difference in the percentage of error between these three wards ($p < 0.05$), but this was clinically significant. We observed total insignificant. Out of 25 nurses subjected to observations, no significant relationship was found between the nurses' characteristics and their rates of errors ($p > 0.05$). Twelve nurses attended in both shifts; 4 attended only in the morning shift, and 8 nurses attended only in the evening shift. Only one nurse was observed 5 times because she left for her maternity leave. From 525 observations, 322 cases happened in morning shifts, where 1011 errors were detected (out of 3328 potential errors (SPE) in the morning shift). A total of 557 errors were detected in the evening shift (out of 2019 potential errors (SPE) in this shift). The rate of error was higher in the morning shift (30.3%) than in the evening shift (27.5%).

Table 3. Percentage of errors in prescribed IV medications (sorted in descending order by number of errors).

| Observation stage | Name of drug | Number of observations | Number of errors | Proportional Error Ratio * (%) | SPE¥ | CTOE£ |
|---|---|---|---|---|---|---|
| **In Total** | Imipenem+cilastatin | 100 | **407** | **25.9** | 1218 | 33.4 |
| | Calcium gluconate | 54 | **168** | **10.6** | 496 | 33.8 |
| | Vancomycin | 42 | **145** | **9.2** | 496 | 29.2 |
| | Cyclosporine | 32 | **131** | **8.3** | 292 | 44.8 |
| | Pantoprazole | 29 | **127** | **5.4** | 266 | 31.9 |
| **Preparation stage** | Imipenem+cilastatin | 100 | **325** | **34.4** | 900 | 36.1 |
| | Vancomycin | 53 | **92** | **5.3** | 106 | 47.1 |
| | Cyclosporine | 32 | **64** | **6.7** | 160 | 40.0 |
| | Pantoprazole | 31 | **91** | **9.6** | 279 | 32.6 |
| | Calcium gluconate | 21 | **83** | **3.9** | 105 | 35.2 |
| **Administration stage** | Calcium gluconate | 54 | **85** | **13.6** | 225 | 37.7 |
| | Imipenem+cilastatin | 100 | **82** | **8.4** | 127 | 41.7 |
| | Cyclosporine | 32 | **67** | **10.7** | 132 | 50.7 |
| | Filgrastim | 29 | **61** | **9.7** | 121 | 50.4 |
| | Metronidazole | 53 | **57** | **9.1** | 176 | 32.3 |

* Total number of errors = 1568.

¥ SPE = Sum of potential errors for each drug. SPE is different for each drug.

£ CTOE = corrected total opportunities for error (Number of errors/SPE)

**Table 4. Percentage of errors in prescribed IV medications (sorted in descending order by CTOE).**

| Observation stage | Name of drug | Number of observations | Number of errors | Proportional Error Ratio *(%) | SPE[¥] | CTOE[£] |
|---|---|---|---|---|---|---|
| **In Total** | Cyclosporine | 32 | 131 | 8.3 | 292 | **44.8** |
| | Vitamin k | 7 | 29 | 1.8 | 71 | **40.8** |
| | Metronidazole | 53 | 107 | 6.8 | 282 | **37.9** |
| | Voriconazole | 2 | 8 | 0.5 | 23 | **34.7** |
| | Calcium gluconate | 54 | 168 | 10.6 | 496 | **33.8** |
| **Preparation stage** | Granisetrone | 24 | 45 | 4.7 | 70 | **64.0** |
| | Metronidazole | 53 | 50 | 5.3 | 106 | **47.1** |
| | Amikacin | 2 | 4 | 0.4 | 10 | **40.0** |
| | Cyclosporine | 32 | 64 | 6.7 | 160 | **40.0** |
| | Imipenem+cilastatin | 100 | 325 | 34.4 | 900 | **36.1** |
| **Administration stage** | Vit k | 7 | 22 | 3.5 | 36 | **61.1** |
| | Voriconazole | 2 | 4 | 0.6 | 7 | **57.1** |
| | Cyclosporine | 32 | 67 | 10.7 | 132 | **50.7** |
| | Filgrastim | 29 | 61 | 9.7 | 121 | **50.4** |
| | Ciprofloxacin | 3 | 5 | 0.8 | 10 | **50.0** |

* Total number of errors = 1568.

[¥] SPE = Sum of potential errors for each drug. SPE is different for each drug. [£] CTOE = corrected total opportunities for error (Number of errors/SPE)

## Discussion

Medication regimens used in BMT wards are complex, including antineoplastic and immuno-suppressant drugs that have a narrow therapeutic index and can be toxic even at therapeutic ranges. It is notable that BMT patients are particularly vulnerable to drug toxicities due to their special condition [30, 31]. IV therapy, as well, is particularly a complicated and error-prone procedure due to the multi-step preparation and sometimes prolonged administration process [32, 33]. Compared to other routes of drug administration, IV therapy is associated with a greater likelihood of medication errors [33]. Error-checking at any point in this process can reduce the probability of error by one-third. Therefore, it is important to understand where and how to check for the frequency of these errors during the preparation or administration of IV medications in patients undergoing HSCT [16].

We evaluated the frequency of various types of MEs in three BMT wards at a referral hospital through direct observation, which is considered the most efficient and accurate method for detecting MEs [13]. We have tried to find and establish a denominator in reporting ME rates, so that we can overcome the underestimation of MEs, misunderstanding of MEs and the share of responsible medications. Therefore we calculated CTOE, representing the number of potential errors for each drug that could have been converted into a true error based on the drug's characteristics. This is important because not all drugs are at the same risk for all types of errors.

Our study estimated a CTOE of 29.3% for the preparation and administration of IV drugs (1568 errors out of 5347 SPEs). The majority of errors in our study occurred during the administration stage due to the wrong administration rate. We found that the most frequent drug subjected to ME was cyclosporine. By comparing the TOE, CTOE, and Proportional Error Ratio estimated in our study, we could understand that how many times an error is detected for a drug might not be an accurate representative of the rate at which a drug was erroneously prepared or administered, even if the drug was only used a few times.

Since we found no other direct observations in this setting, it was challenging to compare our results with the literature. However, a systematic review published in 2019 on MEs in the

BMT setting, including 11 studies (eight case reports and three original studies) reported that most medication errors occurred during the administration and/or prescription stages [30].

Chemotherapy wards are believed to resemble BMT wards in many ways, yet there are a limited number of studies on this topic, making it challenging for us to compare our results with medication errors in the chemotherapy inpatient setting. Moreover, most available studies did not use direct observation or focus on MEs related to IV medications [8, 31, 34]. Nevertheless, we here provided a comprehensive discussion on the subject matter despite the limitations in the relevant literature.

## Corrected total opportunity for errors and potential errors

Here, we first comprehensively explained our main finding with regard to CTOE. The Total Opportunities for Errors (TOE) is a routine indicator used to evaluate MEs. It is calculated as the number of errors per sum of all observed doses administered plus omitted medications. However, TOE is a very general fraction that does not provide specific guidance or details, so it cannot be specifically used for every drug or situation. To address this issue, we proposed a new denominator called the Sum of Potential Errors (SPEs) and a new fraction called Corrected Total Opportunity for Errors (CTOE). These new indicators could offer a more accurate estimation of total MEs for each stage and for each drug. CTOE is calculated as the number of errors for each drug that could have been turned into an actual error based on the drug's characteristics, reflecting that not all drugs are at the same risk for all types of errors.

By comparing TOE, CTOE, and the Proportional Error Ratio (Tables 3 and 4), we could understand that the frequency of errors for a drug might not be an accurate representative of the rate at which a drug was erroneously prepared or administered, even if the drug was only used a few times. For example, in this study, imipenem + cilastatin acquired more errors because of more observations (100 times), with one-third of errors happening during preparation and/or administration (CTOE = 33.4). On the other hand, voriconazole was only observed twice, for which eight errors were detected (CTOE = 34.7), indicating almost the same share compared to imipenem + cilastatin in the occurrence of MEs in the BMT ward.

In a systematic review by MacLeod *et al.*, five important parameters were suggested for selecting denominators in ME studies [6]. These included a usable guide and example, clear inclusion and exclusion criteria for entering doses into the denominator, types of orders, the dose-denominator relationship (if the denominator is not the dose), and the number of doses excluded from the study. Accordingly [6], we identified specific error-prone points during drug administration/preparation that needed to be addressed urgently using CTOE.

Continuing on, we compare our results with findings obtained from direct observation studies that evaluated MEs in IV preparation and administration steps, regardless of their settings and whether they considered other routes or steps.

## Medication error rate

Most errors in our study occurred during the medication administration step (60.2% of total errors). McDowell *et al.* conducted a systematic review and Bayesian analysis of nine studies on the preparation and administration of IV medications and stated that the probability of at least one error occurring in this process was as high as 73% [32]. A similar result was obtained by another systematic review of administration errors in IV medicines, reporting an error rate of 78.6% [33]. In another systematic review of IV medications in the UK, administration errors were higher than preparation errors (32.1% vs. 8.6%) [7], which was similar to our study. We found that 98.8% of the doses administered in three BMT wards in our study carried at least one error. Ong *et al.* reported in their study that 97.7% of the IV doses directly observed by

them had preparation and administration errors [35]. In general, our findings were consistent with those of other studies. However, when analyzing our data in detail, we faced variabilities in ME rates between and within both denominators and numerators [33]. These variations may be due to different definitions of MEs, making it difficult to compare results obtained even by the same methodology; for example, the inclusion of poor aseptic method or wrong administration time as MEs. Direct observation offers the highest accuracy in detecting MEs, and it is well established that it can identify more errors compared to self-reporting and incident reporting, which seriously underestimate the prevalence of MEs [7, 33]. Moreover, ME rates can significantly vary when using different types of numerators and denominators, for example, the number of doses with one or more MEs, the total number of errors, total opportunities for error, and/or self-modified definitions.

It seems that an increase in the number of observed errors, such as in our study, can increase the number of total errors detected because of a higher probability of error detection. However, in a systematic review by Lisby *et al.*, this assumption was completely rejected, clarifying that error detection was largely dependent on the sensitivity of the error detection method and the type of error [12], for which a detailed discussion has been provided below.

## Preparation errors

The most frequent preparation errors observed in the present study were exposure to light before use, using the wrong volume of the reconstitution solvent, and the wrong volume of the dilution solvent. Our findings were consistent with those of other studies, except for the item of exposure to light before use. Errors in the reconstitution [32, 36, 37] and dilution [7, 32, 35] steps were reported as the most frequent IV preparation errors. As declared by McDowell *et al.*, MEs were most likely to happen in the reconstitution step. Errors in the reconstitution or dilution process can inflict serious harm to patients by reducing the drug's solubility, stability, and activity, as well as the appearance of aggregations in the solution and drug sedimentation [35]. Exposure to light before use can disrupt the drug's stability by accelerating oxidation reactions [38]. Under controlled conditions and using a visible-light-driven photocatalyst, the degradation rates of the antineoplastic agents of cyclophosphamide, paclitaxel, MTX, irinotecan, cytarabine, and 5-fluorouracil were determined as 37.7%, >99%, 57.1%, 54.6%, 69.5%, and 36.3%, respectively [39].

We found no mention of the error "exposure to light before use" (for drugs such as methotrexate, which should be protected [38]) in other studies, which may be due to our very narrow classification of errors. Some studies have even considered the entire preparation process as a single step without further mentioning specific errors [33, 35]. We cannot emphasize more the importance of every detailed step in the medication preparation/administration process in the BMT setting. Based on a review by Lermontov *et al.*, BMT patients were 4 to 7 times more likely to die if they experienced adverse events pertaining to MEs. Even if their vital functions remain stable, some patients may have to change their conditioning regimens or be readmitted [30]. Therefore, we here examined the details of each step for individual drugs used.

## Administration errors

Several systematic reviews and experimental studies on the administration of IV medications have noted the most frequent errors as follows: wrong time [7, 33, 35], wrong administration rate [33, 36], wrong dose [33, 37], and wrong administration techniques. In our study, the three most frequent errors in this phase were wrong administration rate, omission of monitoring during or after administration, and exposure to light during infusion. Our findings agree with those of previous studies. Exposure to light during infusion is an error pertaining to the

administration technique. On the other hand, monitoring during or after administration might have been merged into more general categories in other studies. As we mentioned earlier, we here prepared a detailed classification of errors in each step compatible with the BMT setting.

In the present study, the wrong administration rate for an IV drug mostly refers to injections at faster than recommended rates. Without considering the drug's characteristics, fast administration can be associated with pain, phlebitis, and loss of cannula patency [36]. This error has been attributed to nurses' insufficient time and high workload. In fact, nurses have to faster administer drugs to all patients at the same round or accept the risk of the wrong time error [35].

### Error-prone drugs

We found that the most frequent drug involved in MEs was cyclosporine [44.8%], which according to a review conducted by Lermontov *et al.*, this medication was also the most commonly involved drug in MEs in the BMT setting [30].

The error rate of 29.3% for the preparation/administration of IV drugs is alarmingly high in the context of a critical setting such as BMT. This would mean more than one error when it is calculated based on TOE [2.98]. In fact, considering the number of errors per drug observed and per patient undergoing HSCT, we encountered about 3 errors per observed drug and 14 errors per patient, indicating the extremely high frequency of MEs and highlighting the urgent need for implementing preventive measures.

Based on our findings, we recommend conducting further studies to evaluate the applicability of CTOE for different settings and different types of medications. Additionally, we suggest implementing preventive measures in BMT wards to reduce MEs and assessing their effectiveness, feasibility, and costs based on CTOE.

### Limitations

Our study has some limitations. First, we only assessed process errors and did not evaluate their clinical outcomes or contributing factors. The use of convenient sampling for selecting nurses was another limitation, which was due to the complexity and inflexibility of the BMT setting. Furthermore, there are some potential drawbacks to direct observation, such as observer bias or fatigue, and potential subjective interpretations by observers. However, direct observation remains the most accurate method for detecting MEs. In our study, the main parameters, such as SPE and CTOE, were almost independent of observer bias.

### Conclusion

Our study provided a detailed description of the prevalence of MEs in BMT wards using the most accurate detection method (i.e., direct observation) and by defining a new denominator based on a comprehensive checklist. Our findings can help plan targeted preventive measures specifically designed for the BMT setting and investigate the effectiveness, feasibility, and costs of these measures. Future studies can use our method to identify MEs during preparation and administration in different settings.

### Supporting information

**S1 File. Checklist is the S1 File title.**
(DOCX)

**S2 File. Dataset in full English is the S2 File title.**
(XLSX)

## Acknowledgments

We would like to appreciate BMT wards, the Hospital nursing staff for their effort and collaboration.

## Author Contributions

**Conceptualization:** Ava Mansouri, Kazem Heidari, Molouk Hadjibabaie.

**Data curation:** Kiana Moazzeni, Maryam Valeh.

**Formal analysis:** Ava Mansouri, Kazem Heidari.

**Funding acquisition:** Molouk Hadjibabaie.

**Investigation:** Ava Mansouri, Kiana Moazzeni, Maryam Valeh, Kazem Heidari.

**Methodology:** Maryam Valeh, Kazem Heidari.

**Project administration:** Ava Mansouri, Molouk Hadjibabaie.

**Supervision:** Ava Mansouri, Molouk Hadjibabaie.

**Validation:** Ava Mansouri, Maryam Valeh, Kazem Heidari, Molouk Hadjibabaie.

**Writing – original draft:** Ava Mansouri.

**Writing – review & editing:** Kazem Heidari, Molouk Hadjibabaie.

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
