## [Decision Letter · Decision Letter 0]

30 Aug 2023

PONE-D-23-13349How to get over with medication errors underestimation? Improving indices of medication errors with focus on intravenous medications in hematopoietic stem cell transplantation setting; a direct observation study.PLOS ONE

Dear Dr. Mansouri,

Thank you for submitting your manuscript to PLOS ONE. After careful consideration, we feel that it has merit but does not fully meet PLOS ONE’s publication criteria as it currently stands. Therefore, we invite you to submit a revised version of the manuscript that addresses the points raised during the review process.

We look forward to receiving your revised manuscript.

Kind regards,

Ali Amanati

Academic Editor

PLOS ONE

Journal Requirements:

"This study was supported by the Research Council of Tehran University of Medical Sciences [grant number: 22297-156-01-92]. "       

Additional Editor Comments:

Dear authors

Your manuscript [PONE-D-23-13349] has passed the review stage and is ready for ‎revision. ‎

Editorial comments

To ensure the Editor and Reviewers can recommend that your revised manuscript be ‎accepted, ‎‎please pay careful attention to each comment posted underneath ‎this email. This way we ‎can ‎avoid future clarifications and revisions, moving swiftly to ‎a decision.‎

‎1. Please provide a point-by-point response to the Editor and reviewer's comments

‎2. Please highlight all the amends on your manuscript with yellow color

‎3. ‎Some grammatical and spacing errors need to be revised throughout the ‎manuscript

Reviewers' comments:

Reviewer's Responses to Questions

**Comments to the Author**

1. Is the manuscript technically sound, and do the data support the conclusions?

Reviewer #1: Partly

Reviewer #2: Partly

Reviewer #3: Yes

Reviewer #4: Yes

Reviewer #5: Yes

Reviewer #6: Yes

2. Has the statistical analysis been performed appropriately and rigorously? 

Reviewer #1: Yes

Reviewer #2: No

Reviewer #3: Yes

Reviewer #4: Yes

Reviewer #5: Yes

Reviewer #6: N/A

3. Have the authors made all data underlying the findings in their manuscript fully available?

Reviewer #1: Yes

Reviewer #2: Yes

Reviewer #3: Yes

Reviewer #4: Yes

Reviewer #5: Yes

Reviewer #6: Yes

4. Is the manuscript presented in an intelligible fashion and written in standard English?

Reviewer #1: Yes

Reviewer #2: Yes

Reviewer #3: Yes

Reviewer #4: Yes

Reviewer #5: Yes

Reviewer #6: Yes

5. Review Comments to the Author

Reviewer #1: Dear authors I would like to thank you for the nice and very important topic you had explored but i have the following comments for further improvement:

1- The abstract should not contain abbreviations so please remove all and advice to read journal requirements for submission.

2- The abstract should be divided under subtitles (introduction, methods, results and discussion) so pleas restructure your abstract.

3- The references at the end of cited sentences should be inside brackets [ ] not ( ), so please modify all.

4- I have one major concern and really you need to explain that very well in your manuscript, how the results were validated? I mean did you validate the observations that were conducted by one person which are more prone to bias, if yes how ? if not why you didn't that and how you will overcome this major issue?

Reviewer #2: There are a few points that are necessary to be considered

1. This study is cross-sectional observational study, while the study was carried out between 2014-2015

2. The data were collected from three wards, related to Teaching hospital. The possibility of bias is high

3. The results were presented as descriptive analysis, while the inferential analysis between the three wards were missed

4. It is difficult to get a conclusion fro data obtained from specific wards

5. Check the references: typing errors

6. This study that was done in 2014-2015 will not add new information in 2023

Reviewer #3: The manuscript technically sound, and the data support the conclusions. The statistical analysis has been performed appropriately. The conclusions drawn appropriately based on the data presented. The manuscript presented in clear and comprehensible fashion

Reviewer #4: After reviewing the current manuscript:

1- The paper is well written.

2- All references are updated.

3- The authors discussed the main idea successfully.

I recommend the acceptance of the paper in its current form.

Reviewer #5: Major Points and Feedback:

Data Collection Tools and Analysis:

The use of SPSS-21 for data analysis is well-suited for this type of study. The choice of using Excel for data input is reasonable; however, more details on how data were cleaned and processed before analysis would be useful.

Descriptive Statistics:

The presentation of data through descriptive statistics such as mean, standard deviation, frequency, and percentage is well executed. This aids in understanding the spread and central tendencies of the data.

Total Opportunities for Errors (TOE) & Corrected Total Opportunities for Errors (CTOE):

The distinction between TOE and CTOE is a crucial one and offers a novel approach. The authors should consider focusing more on explaining the real-world implications of this distinction. How does CTOE improve upon existing error rate measurements?

Results Section:

The results are detailed and cover a broad range of observed data. However, it may be helpful to provide more context or a brief summary before diving into specific numbers.

It would be beneficial to discuss why certain drugs, like Imipenem+cilastatin, had high error rates. Are these errors due to the complexity of the drug's administration, or are there other factors at play?

Tables:

Tables are well-presented, offering a breakdown of errors, observations, and characteristics. However, using clearer labels and perhaps footnotes to explain terms such as SPE, CTOE, etc., directly in the table can enhance readability.

Potential Limitations:

The study observes only 25 nurses across three wards. A discussion on the representativeness of the sample and any potential biases in observation would be beneficial.

The paper could delve deeper into the implications of the fact that all observed nurses were female and held a Bachelor of Nursing degree.

Recommendations for Improvement:

The introduction of the concept of CTOE should be supported by further literature. If this is a novel concept introduced by the authors, a more in-depth rationale behind its introduction would be beneficial.

A discussion or conclusion section summarizing key findings and their implications would add value. Recommendations for reducing error rates based on the findings would also be insightful.

It might be useful to compare the findings with similar studies (if any) to provide a benchmark or context to the observed error rates.

Conclusion:

The paper presents a comprehensive analysis of medication errors in drug preparation and administration, introducing novel concepts to enhance understanding. While the methodology and presentation of results are rigorous, there are areas for improvement in explanation, context, and discussion. With further refinement, this paper can serve as a significant contribution to the field of medication error research.

Reviewer #6: I only have one major comment about the study. The authors mentioned that a final year pharmacy student served as the observer, who compared their notes with the medical chart, order sheet, and nurses' notes every day. Is it possible that the observer could make errors and have the authors conducted any reliability analysis?

6. PLOS authors have the option to publish the peer review history of their article (what does this mean?). If published, this will include your full peer review and any attached files.

Reviewer #1: **Yes: **Bayan Ababneh

Reviewer #2: **Yes: **Marwan S.M. Al-Nimer

Reviewer #3: **Yes: **Thaer Abdelghani

Reviewer #4: No

Reviewer #5: **Yes: **Gustavo A Fernandez

Reviewer #6: No

---

## [Author Response · Author response to Decision Letter 0]

7 Jan 2024

Meeting the Journal Requirements based on editors comment:

1. We have corrected the manuscript based on PLOS ONE's style requirements. 

2. It was asked to provide additional details regarding participant consent. We have specified in the ethics statement in the methods, that we obtained the written consent form from patients. We will provide this information as well in online submission information. We also have excluded the minors from our study.

 3. We have to state that “The funders had no role in study design, data collection and analysis, decision to publish, or preparation of the manuscript." We have added this statement also to our manuscript

 4. The ethics statement in our manuscript is in under “Compliance with Ethical Standards” in the method section as you have requested 

5. We have provided the Excel worksheet of our data in anonymized form and excluded those sheets and parts which were not necessary or included the names’ codes. We will upload it as a supplementary file. 

The response to reviewers’ comments;

Thank you for giving us the opportunity to submit a revised draft of the manuscript “How to get over with medication errors underestimation? Improving indices of medication errors with focus on intravenous medications in hematopoietic stem cell transplantation setting; a direct observation study.” for publication in the PLOS ONE journal.

We appreciate the time and effort that you and the reviewers dedicated to review our manuscript. We are grateful for the insightful comments on and valuable improvements to our paper. We have considered carefully the comments and addressed most of the suggestions. 

We have highlighted the changes within the manuscript with track change and made a point-by-point response to the reviewers’ comments and concerns. 

Reviewer #1: Dear authors I would like to thank you for the nice and very important topic you had explored but i have the following comments for further improvement: 

Authors answer; thank you, we are so grateful for your interest in our manuscript and your valuable comments

1- The abstract should not contain abbreviations so please remove all and advice to read journal requirements for submission. 

Authors answer; the Changes are applied as reviewer suggested

2- The abstract should be divided under subtitles (introduction, methods, results and discussion) so pleas restructure your abstract. 

Authors answer; the Changes are applied as reviewer suggested

3- The references at the end of cited sentences should be inside brackets [ ] not ( ), so please modify all.

Authors answer; the Changes are applied Changes are applied as reviewer suggested

4- I have one major concern and really you need to explain that very well in your manuscript, how the results were validated? I mean did you validate the observations that were conducted by one person which are more prone to bias, if yes how ? if not why you didn't that and how you will overcome this major issue? 

Authors answer; Thank you for pointing out this important issue. We are totally aware of the significance of the concern you have raised. Actually we had conducted a first round study, following all the steps and under the same conditions as the main study. Unfortunately we have thought that this might not be essential due to amount of the details. Therefore, here we will explain the whole process and will also include it in the method due to yours and other reviewers’ reasonable concern;

Initially, the observer underwent training by the clinical pharmacist professor, who oversees all three BMT wards. Subsequently, the observer utilized the prepared checklist and conducted a pilot study in the relevant wards. The observer diligently documented information from patients' charts, recorded her observations, and completed the checklists. Following this, she presented the gathered information to the study supervisor, a Pharm.D pharmacist, who independently filled out additional checklists based on the observer's findings.

Subsequently, a comprehensive comparison of the completed checklists was undertaken by the team, comprising the clinical pharmacist professor in charge, the responsible oncologist of the wards, the study supervisor, and the observer. Any discrepancies in the errors detected by the observer and the supervisor were categorized as true, false positive, and false negative. They also deliberated on the classification of the errors. Subsequently, necessary adjustments were made to the observation process and error detection. Additionally, during the primary study phase, some observations were randomly verified by the supervisor.

Reviewer #2: There are a few points that are necessary to be considered

1. This study is cross-sectional observational study, while the study was carried out between 2014-2015; 

Authors answer; The study was cross-sectional lasting from 2014 to 2015, and there was no follow up afterward. 

2. The data were collected from three wards, related to Teaching hospital. The possibility of bias is high

Authors answer: We agree with the reviewer’s assessment. Thank you for your concern and pointing out this important issue, here we have your answer as we explained in response to other reviewers; 

Actually, we conducted a preliminary study, following all the steps and under the same conditions as the main study. Regarding your concern, we explained the whole process here and in the methods section.

Initially, the observer underwent training by the clinical pharmacist professor, who oversees all three BMT wards. Subsequently, the observer utilized the prepared checklist and conducted a pilot study in the relevant wards. The observer diligently documented information from patients' charts, recorded her observations, and completed the checklists. Following this, she presented the gathered information to the study supervisor, a Pharm.D pharmacist, who independently filled out additional checklists based on the observer's findings.

Subsequently, a comprehensive comparison of the completed checklists was undertaken by the team, comprising the clinical pharmacist professor in charge, the responsible oncologist of the wards, the study supervisor, and the observer. Any discrepancies in the errors detected by the observer and the supervisor were categorized as true, false positive, and false negative. They also deliberated on the classification of the errors. Subsequently, necessary adjustments were made to the observation process and error detection. Additionally, during the primary study phase, some observations were randomly verified by the supervisor.

3. The results were presented as descriptive analysis, while the inferential analysis between the three wards were missed

Authors answer; thank you for pointing this issue out. 

We have conducted a comparison between wards. Although we will include the results in the relevant section, we will also provide the explanation here. The characteristics of the wards were the same; they all had 9 beds with a nurse-patient ratio of 1 to 4 or 1 to 5. We also analyzed the characteristics of the nurses and could not find any significant differences between their characteristics and the medication errors they were involved in. Although we found that error rates varied significantly between wards, the close range of their CTOEs (24 to 31) made it clinically unimportant.

4. It is difficult to get a conclusion from data obtained from specific wards

Authors answer; thank you for sharing your concern. However, as we mentioned before, our primary objective was to identify the most effective approach for reporting medication errors that occur in the BMT wards of a hospital, specifically for patients undergoing HSCT. We aimed to achieve this through a well-defined, precise detection method, using a denominator and checklist. This reporting method offers superior clarity and provides more detailed information through its indicators. Consequently, it serves as a valuable guide for developing targeted preventive measures and assessing their effectiveness. It is crucial to investigate the applicability of this method in other settings to fully understand its significance and relevance. 

5. Check the references: typing errors

Authors answer; Changes are applied as reviewer suggested

6. This study that was done in 2014-2015 will not add new information in 2023

Author response: While we appreciate the reviewer’s feedback, we respectfully disagree. We think this study makes a valuable contribution to the field because the issue that we raised and took action to solve is still an important issue as well. Because the reporting of medication error articles is still done in an old-fashioned way and total opportunities for error (TOE) is still common and established. Our goal is to review TOE, which is still routine in 2023.

Reviewer #3: The manuscript technically sound, and the data support the conclusions. The statistical analysis has been performed appropriately. The conclusions drawn appropriately based on the data presented. The manuscript presented in clear and comprehensible fashion

Author response: thank you, we are so grateful for your time and effort. We are also very pleased that you find the manuscript interesting as the way it is. 

Reviewer #4: After reviewing the current manuscript:

1- The paper is well written.

2- All references are updated.

3- The authors discussed the main idea successfully.

I recommend the acceptance of the paper in its current form.

Author response: We are so grateful for your valuable time and assessment and we are so pleased that you already find the manuscript interesting as it is. 

Reviewer #5: Major Points and Feedback:

• Data Collection Tools and Analysis:

The use of SPSS-21 for data analysis is well-suited for this type of study. The choice of using Excel for data input is reasonable; however, more details on how data were cleaned and processed before analysis would be useful.

Author response: Thank you for this suggestion. The observer recorded the data and error details from the checklists into Excel after each observation. The columns included the following information: ward name, nurse code, observation number, date, morning/evening shifts, drug generic name, drug brand name, dosage forms (powder or solution), dosage unit, number of administrations in one day (QD, BD, TDS, QOD). The subsequent columns were dedicated to error types, with the following coding: 1 - correct checkpoint, 2 - incorrect checkpoint (error). In cases where errors occurred, such as monitoring or exposure to light during administration, there were three possible conditions: 1 - not relevant for that drug, 2 - should be considered and done correctly, 3 - should be considered and not done (error). Finally, the demographic data of the nurses for each observed drug were entered into the last columns. I hope this explanation is satisfactory and provides sufficient clarity.

• Descriptive Statistics:

The presentation of data through descriptive statistics such as mean, standard deviation, frequency, and percentage is well executed. This aids in understanding the spread and central tendencies of the data.

Total Opportunities for Errors (TOE) & Corrected Total Opportunities for Errors (CTOE):

The distinction between TOE and CTOE is a crucial one and offers a novel approach. The authors should consider focusing more on explaining the real-world implications of this distinction. How does CTOE improve upon existing error rate measurements? 

Author response: thank you for pointing this topic out. CTOE can be accounted as standardized error rate. Since the drugs have different characteristics and their share in total error occurrence is absolutely different. CTOE can reveal the real causes and largest share of errors in the drug utilization process (both in the terms of drugs and neglected stages) 

• Results Section:

The results are detailed and cover a broad range of observed data. However, it may be helpful to provide more context or a brief summary before diving into specific numbers.

Author response: the Changes are applied as reviewer suggested

It would be beneficial to discuss why certain drugs, like Imipenem+cilastatin, had high error rates. Are these errors due to the complexity of the drug's administration, or are there other factors at play?

Author response: Thank you for pointing this out. While we acknowledge the importance of this consideration, it is beyond the scope of this manuscript as our study design and method are not suitable for determining the causes of medication errors. We are only able to present which medications have what types of errors and their prevalence. Additionally, imipenem+cilastain had a high number of errors, but it was not the most erroneous drug overall based on CTOE. As we previously reported, cyclosporine was the most erroneous medication in total based on CTOE, meaning it had the highest number of errors in the BMT wards, which we have discussed in the erroneous drugs section of our discussion.

Tables:

Tables are well-presented, offering a breakdown of errors, observations, and characteristics. However, using clearer labels and perhaps footnotes to explain terms such as SPE, CTOE, etc., directly in the table can enhance readability. 

Author response; We have tried to make it more clear and I hope your request is resolved.

Potential Limitations:

The study observes only 25 nurses across three wards. A discussion on the representativeness of the sample and any potential biases in observation would be beneficial.

Authors answer: We totally agree with the reviewer’s assessment. Thank you for your concern and pointing out this important issue, here we have your answer as we explained in response to other reviewers earlier; 

Actually we had conducted a first round study, following all the steps and under the same conditions as the main study. Unfortunately we have thought that this might not be essential due to amount of the details. Therefore, here we will explain the whole process and will summarize it in the method due to yours and other reviewers’ reasonable concern;

At first, the observer was trained by the clinical pharmacist professor who is in charge of all the tree BMT wards. Then the observer referred to the related wards with the prepared checklist and did a test study in the field. The observer has recorded all the information from patients’ chart as well as all of her observations and filled out the checklists. Afterward she delivered the gathered information to the study supervisor (a Pharm.D pharmacist) and the supervisor also filled the checklists dependently based on the information that observer provided. Afterward a thorough comparison between the completed checklists was done by the team of the in charge clinical pharmacist professor, the responsible oncologist of the wards, the study supervisor and the observer. The diversity between the observer and the supervisor regarding the detected errors were categorized as; true, false positive and false negative. In addition they have discussed where the errors belong (in which categories). The necessary modifications made to the observation process and errors detection. During the main study phase some of observations were randomly checked by the supervisor as well.

 We have also done the comparison between wards, we now also add it in the results part. The ward characteristics were the same; they all have 9 beds with nurse patient ratio 1 to 4 or 1 to 5. We also analysis the characteristics of nurses and we could not find any significant differences between their characteristics and medication errors they have engaged. Although we have found that error rates were significantly different between wards but since their CTOEs are close and ranged between from 24 to 31, it was not clinically important.

The paper could delve deeper into the implications of the fact that all observed nurses were female and held a Bachelor of Nursing degree.

Author response: Thank you for mentioning this point. Although we agree that this can be an important issue, but it is beyond the scope of this manuscript , since talking about these characteristics that you are pointing out, is lying in the concept of causes of errors. As far as it was within the scope of our study, the

---

## [Decision Letter · Decision Letter 1]

11 Apr 2024

PONE-D-23-13349R1How to get over with medication errors underestimation? Improving indices of medication errors with focus on intravenous medications in hematopoietic stem cell transplantation setting; a direct observation study.PLOS ONE

Dear Dr. Mansouri,

Thank you for submitting your manuscript to PLOS ONE. After careful consideration, we feel that it has merit but does not fully meet PLOS ONE’s publication criteria as it currently stands. Therefore, we invite you to submit a revised version of the manuscript that addresses the points raised during the review process.

We look forward to receiving your revised manuscript.

Kind regards,

*
**Ali Amanati**
*

**Academic Editor**

*
**PLOS ONE**
*

Additional Editor Comments:

**Dear authors**,

‎New comments were posted by the invited reviewers. So, the manuscripts require a ‎round of revision.‎ Please provide a point-by-point response to the reviewer ‎comments and highlight all the ‎amends on your manuscript with yellow color.‎

It should be mentioned that some invited reviewers declined to participate in the new peer review round so additional reviewers were invited.

Yours

Reviewers' comments:

Reviewer's Responses to Questions

**Comments to the Author**

1. If the authors have adequately addressed your comments raised in a previous round of review and you feel that this manuscript is now acceptable for publication, you may indicate that here to bypass the “Comments to the Author” section, enter your conflict of interest statement in the “Confidential to Editor” section, and submit your "Accept" recommendation.

Reviewer #8: All comments have been addressed

Reviewer #9: (No Response)

Reviewer #10: (No Response)

2. Is the manuscript technically sound, and do the data support the conclusions?

Reviewer #8: Partly

Reviewer #9: Partly

Reviewer #10: Yes

3. Has the statistical analysis been performed appropriately and rigorously? 

Reviewer #8: I Don't Know

Reviewer #9: No

Reviewer #10: Yes

4. Have the authors made all data underlying the findings in their manuscript fully available?

Reviewer #8: Yes

Reviewer #9: Yes

Reviewer #10: Yes

5. Is the manuscript presented in an intelligible fashion and written in standard English?

Reviewer #8: No

Reviewer #9: No

Reviewer #10: Yes

6. Review Comments to the Author

Reviewer #8: Dear authors

I appreciate the efforts in revising the manuscript as recommended by the reviewers.

I strongly recommend the manuscript is given to a native English user/speaker to correct all grammatical errors and break down winding sentences for conciseness and easier comprehension.

Reviewer #9: Thank you for this clinical relevant study. I have organised my feedback into major and minor comments as below

Major

Generic

Please avoid subjective language, such as "most defined," "great help," and "poor." English editing before resubmitting is recommended.

Abstract

1 The definition of CTOE is provided, but the terms TOE and Proportional Error Ratio need clarification as well.

2 The presentation of figures such as 2.98 or 298% is unclear without context. Additionally, sometimes the inconsistent mention of CTOE and SPE and the absence of data for cyclosporine are confusing. I would recommend only using the most important indicators in the abstract.

Introduction

1 The mention of HSCT seems abrupt. Please restructure the paragraph to highlight the differences between HSCT settings and general settings.

2 It is unclear what is being compared when mentioning the benefits of preventive ME interventions. Please specify.

3 The potential barriers in epidemiology studies within this field require clarification to strengthen the introduction.

4 The calculation of 16-365 times is unclear, as the reference does not directly mention these figures. Please also clarify the difference between these methods.

5 The significance of denominators is mentioned, but not in the context of the study's objectives.

6 A more concise introduction is recommended for clarity.

Method

1 Briefly describe the "disguised direct observation method" based on Barker and McConnell's practice, including its performance in prior research.

2 The rationale for selecting the 32 medications observed should be clarified.

3 Details on how random verification was performed are needed.

4 The reasoning behind the relative error checkpoints for each drug could be elaborated in the Supplement.

Results

1 Table 1's format is unclear, particularly regarding the overall characteristics and the association between Ward A and nursing errors.

2 The definition regarding error rates per SPE belongs in the Methods section. Please define these indicators clearly and consistently in the Method and straight use the term, for example, CTOE in the Result section.

3 The abrupt presentation of nurse characteristics and error differences between wards needs to be reorganised.

Discussion

1 Restructure the Discussion to outline main findings, compare with prior research, and discuss strengths and limitations. A significant rewrite is needed for clarity.

2 Please highlight the value of the CTOE at the beginning of the discussion.

3 The argument against direct comparison lacks new insights and needs strengthening.

Minor Comments

Introduction

1 Please keep consistency when discussing dose errors, specifying if it relates to IV medication or dose errors.

2 Once the abbreviation is defined, please use abbreviations consistently.

3 Full terms should be used for clarity, e.g., "denominators of xxx" instead of "denominators."

Method

1 Please use "sex" instead of "gender" for biological distinctions.

2 Please specify which author conducted the observations in each steps.

Results

1 Please clarify the calculation of reported numbers, including numerators and denominators.

Discussion

1 Please improve readability by correctly formatting references, as the current method (e.g., "33 et al.") is incorrect.

Reviewer #10: Thank you for the opportunity to review the article. I'm pleased to inform you that I find the article suitable for publication. The research conducted is rigorous, the findings are significant, and the writing is clear and engaging.

7. PLOS authors have the option to publish the peer review history of their article (what does this mean?). If published, this will include your full peer review and any attached files.

Reviewer #8: No

Reviewer #9: No

Reviewer #10: No

---

## [Author Response · Author response to Decision Letter 1]

9 Jun 2024

The response to reviewers’ comments;

Thank you for giving us the opportunity to submit a revised draft of the manuscript “How to get over with medication errors underestimation? Improving indices of medication errors with focus on intravenous medications in hematopoietic stem cell transplantation setting; a direct observation study.” for publication in the PLOS ONE journal.

We appreciate the time and effort that you and the reviewers dedicated to review our revised manuscript. We are grateful for the insightful comments on and valuable improvements to our paper. We have considered carefully the comments and addressed most of the suggestions. 

We have highlighted the changes within the manuscript with track change and made a point-by-point response to the reviewers’ comments and concerns. Those track changes that are not highlighted are the edited parts by the native English speaker.

Reviewer #8: Dear authors I appreciate the efforts in revising the manuscript as recommended by the reviewers. I strongly recommend the manuscript is given to a native English user/speaker to correct all grammatical errors and break down winding sentences for conciseness and easier comprehension.

Authors answer; we are grateful for the time and effort you spent to review our manuscript. We are so pleased that you are satisfied with the manuscript revision and we have followed your suggestion about the native English editing 

Reviewer #9: Thank you for this clinical relevant study. I have organised my feedback into major and minor comments as below

Major

a. Generic

1- Please avoid subjective language, such as "most defined," "great help," and "poor." English editing before resubmitting is recommended.

Authors answer; Thank you for pointing out this important issue. The manuscript is now edited by a native English speaker.

b. Abstract

1-The definition of CTOE is provided, but the terms TOE and Proportional Error Ratio need clarification as well. 

Authors answer; Thanks to your comment, the definitions are now clarified

2- The presentation of figures such as 2.98 or 298% is unclear without context. Additionally, sometimes the inconsistent mention of CTOE and SPE and the absence of data for cyclosporine are confusing. I would recommend only using the most important indicators in the abstract. 

Authors answer; We have considered your recommendation and the changes are applied as reviewer suggested

c. Introduction

1-The mention of HSCT seems abrupt. Please restructure the paragraph to highlight the differences between HSCT settings and general settings. 

Authors answer; We have revised the text to address your concerns and hope it is now clearer.

2-It is unclear what is being compared when mentioning the benefits of preventive ME interventions. Please specify. 

Authors answer; we are apologizing for the confusion, we have tried to say if we want to demonstrate the benefits of preventive ME interventions, we should be able to compare these interventions. We also have corrected this misleading statement in the manuscript.

3-The potential barriers in epidemiology studies within this field require clarification to strengthen the introduction.

 Authors answer; thank you for pointing out this issue, we have tried to make it understandable

4-The calculation of 16-365 times is unclear, as the reference does not directly mention these figures. 

Authors answer; you are absolutely right, these numbers are not directly reported in the reference “Flynn EA, Barker KN, Pepper GA, Bates DW, Mikeal RL. Comparison of methods for detecting medication errors in 36 hospitals and skilled-nursing facilities. Am J Health Syst Pharm. 2002;59(5): 436-46.”, we calculated them ourselves as we compare number of errors detected by each method. We extract these from the text of the cited article; “Direct observation detected 300 of these errors and 73 false positives, which produced an error rate of 14.6%. For the same doses, chart review detected 17 of the 457 errors and 7 false positives, yielding an error rate of 0.9% (14.6/0.9=16), while incident report review detected only 1 error for an error rate of 0.04% (14.6/0.04=365).”

Please also clarify the difference between these methods.

Authors answer; we tried to explain methods in a concise form as you have asked. 

5-The significance of denominators is mentioned, but not in the context of the study's objectives. I am not sure if I can understand what are you suggesting, 

Authors answer; we appreciate your concern and hopefully we may well make it more understandable with the following explanation; We have tried to find and establish a denominator in reporting ME rates (since the nominators are detected errors in whole or part of it in all MEs studies thus the main problem here is denominators), so that we can overcome the underestimation of MEs, misunderstanding of MEs and the share of responsible medications. That is why we are highlighting denominators role. 

6- A more concise introduction is recommended for clarity. 

Authors answer; we thank you for pointing out this problem, the study is a little complicated that’s why we try to address every necessary things in our manuscript to make it rationality understandable, but we have revised it as much as possible to make it more concise.

d. Method

1 Briefly describe the "disguised direct observation method" based on Barker and McConnell's practice, including its performance in prior research. 

Authors answer; the Changes are applied as reviewer suggested and they are now added to the text

2-The rationale for selecting the 32 medications observed should be clarified. 

Authors answer; as we have mentioned in the method, these 32 IV medications are the most used medications in the HSCT wards reported by the nurses’ supervisor, so we can exclude other not so usual medications which nurses might not be very familiar with their IV administration process. Thanks to your comments, it is now added to the manuscript. 

 3- Details on how random verification was performed are needed. 

Authors answer; the Changes are applied as reviewer suggested and they are now added to the text

4- The reasoning behind the relative error checkpoints for each drug could be elaborated in the Supplement. 

Authors answer; as you have requested, all the details are assigned as supplement to the manuscript and sent to the journal previously

e. Results

1-Table 1's format is unclear, particularly regarding the overall characteristics and the association between Ward A and nursing errors. 

Authors answer; We are sorry for the confusion, this table is only representing the number of each characteristic and there is no association in this table. We have made it clearer in the text and table.

2 The definition regarding error rates per SPE belongs in the Methods section. Please define these indicators clearly and consistently in the Method and straight use the term, for example, CTOE in the Result section. 

Authors answer; thanks to your consideration, we have defined each error rate in method completely individually and we have revised the result as you were suggested

3 The abrupt presentation of nurse characteristics and error differences between wards needs to be reorganised.

Authors answer; we have applied the changes in reports order as reviewer suggested

f. Discussion

1-Restructure the Discussion to outline main findings, compare with prior research, and discuss strengths and limitations. A significant rewrite is needed for clarity.

Authors answer; the changes are applied as reviewer suggested and the discussion has been rewritten

 2-Please highlight the value of the CTOE at the beginning of the discussion. 

Authors answer; thanks for your consideration the CTOE value is now added at the beginning of the discussion

3-The argument against direct comparison lacks new insights and needs strengthening.

Authors answer; thank you for sharing your concern. As we mentioned before, the issue that you are pointing out might be due to limited number of studies available on this topic, as well as the fact that direct observation was not employed as a method in any of these studies, besides the fact that none of the studies specifically focused on evaluating MEs in the IV process or reported errors related to the IV process separately. But we hope this issue is somehow resolved after the discussion is rewritten and edited. 

 Minor Comments

a. Introduction

1-Please keep consistency when discussing dose errors, specifying if it relates to IV medication or dose errors.

Authors answer; the Changes are applied as reviewer suggested 

2-Once the abbreviation is defined, please use abbreviations consistently. 

Authors answer; the corrections are as reviewer suggested

3 Full terms should be used for clarity, e.g., "denominators of xxx" instead of "denominators." 

Authors answer; this is corrected as reviewer suggested

b. Method

1-Please use "sex" instead of "gender" for biological distinctions. 

Authors answer; the Changes are applied as reviewer suggested

2-Please specify which author conducted the observations in each steps. 

Authors answer; the correction is applied as reviewer suggested 

c. Results

1-Please clarify the calculation of reported numbers, including numerators and denominators.

Authors answer; all of the denominators and numerators are defined in the method completely

d. Discussion

1 Please improve readability by correctly formatting references, as the current method (e.g., "33 et al.") is incorrect. 

Authors answer; we are apologizing for this mistake, it is corrected now and the manuscript is given for native English editing 

We would like to thank the referee again for taking the time to review our manuscript.

Reviewer #10: Thank you for the opportunity to review the article. I'm pleased to inform you that I find the article suitable for publication. The research conducted is rigorous, the findings are significant, and the writing is clear and engaging.

Authors answer; thank you for your time and effort to review our manuscript. We are so pleased that you are satisfied with our revision.

---

## [Decision Letter · Decision Letter 2]

8 Jul 2024

How to get over with medication errors underestimation? Improving indices of medication errors with focus on intravenous medications in hematopoietic stem cell transplantation setting; a direct observation study.

PONE-D-23-13349R2

Dear Dr. Ava Mansouri,

We’re pleased to inform you that your manuscript has been judged scientifically suitable for publication and will be formally accepted for publication once it meets all outstanding technical requirements.

Kind regards,

*
**Ali Amanati**
*

**Academic Editor**

*
**PLOS ONE**
*

Additional Editor Comments (optional):

The study offers valuable recommendations for healthcare professionals and institutions to implement strategies that can reduce the occurrence of medication errors, ultimately leading to better outcomes for patients.

Through its detailed analysis and observations, the paper adds to the existing body of knowledge on medication safety in specialized medical settings, offering practical implications for healthcare practice and policy.

The authors have managed to use all the available resources and data to re-shape ‎the manuscript in a ‎manner that is more scientifically sound than previously.‎ So, based on my ‎opinion and the respected ‎reviewers' comments could be published in its current form‎.‎

Yours

Reviewers' comments:

Reviewer's Responses to Questions

**Comments to the Author**

1. If the authors have adequately addressed your comments raised in a previous round of review and you feel that this manuscript is now acceptable for publication, you may indicate that here to bypass the “Comments to the Author” section, enter your conflict of interest statement in the “Confidential to Editor” section, and submit your "Accept" recommendation.

Reviewer #8: All comments have been addressed

2. Is the manuscript technically sound, and do the data support the conclusions?

Reviewer #8: Partly

3. Has the statistical analysis been performed appropriately and rigorously? 

Reviewer #8: N/A

4. Have the authors made all data underlying the findings in their manuscript fully available?

Reviewer #8: Yes

5. Is the manuscript presented in an intelligible fashion and written in standard English?

Reviewer #8: Yes

6. Review Comments to the Author

Reviewer #8: Considering the efforts put in by the authors in addressing our previous comments, I do not have further comments at this point.

7. PLOS authors have the option to publish the peer review history of their article (what does this mean?). If published, this will include your full peer review and any attached files.

Reviewer #8: No

---

## [Editor Report · Acceptance letter]

12 Jul 2024

PONE-D-23-13349R2 

PLOS ONE

Dear Dr. Mansouri, 

I'm pleased to inform you that your manuscript has been deemed suitable for publication in PLOS ONE. Congratulations! Your manuscript is now being handed over to our production team.

Kind regards, 

on behalf of

Professor Ali Amanati 

Academic Editor

PLOS ONE